# Unmet Healthcare Needs and Their Determining Factors among Unwell Migrants: A Comparative Study in Shanghai

**DOI:** 10.3390/ijerph19095499

**Published:** 2022-05-01

**Authors:** Lin Pan, Cong Wang, Xiaolin Cao, Huanhuan Zhu, Li Luo

**Affiliations:** 1School of Public Health, Fudan University, Shanghai 200032, China; 20111020057@fudan.edu.cn (L.P.); 19111020004@fudan.edu.cn (C.W.); 18621600151@163.com (H.Z.); 2Institute of Medical Information, Chinese Academy of Medical Sciences, Beijing 100020, China

**Keywords:** migrants, unmet healthcare needs, Anderson health service utilization model

## Abstract

The purpose of this study was to analyze the health status and unmet healthcare needs, and the impact of related factors, of unwell migrants in Shanghai. A total of 10,938 respondents, including 934 migrants and 10,004 non-migrants, were interviewed in Shanghai’s Sixth Health Service Survey. Descriptive statistics were utilized to present the prevalence of health status and unmet healthcare needs. Binary logistic regression analysis was performed to explore the relationships between predisposing factors, enabling factors, need factors, and health-related behavior and unmet healthcare needs in the Anderson health service utilization model. This study indicated the percentages of migrants having a fair or poor self-evaluated health status (21.09%) and suffering from chronic diseases (72.91%) were lower than those of non-migrants (28.34% and 88.64%, respectively). Migrants had higher percentages of unmet hospitalization needs (88.87%), unmet outpatient care needs (44.43%), and self-medication (23.98%) than those of non-migrants (86.24%, 37.95%, 17.97%, respectively). Migrants enrolled in Urban Employee Basic Medical Insurance were more likely to utilize hospitalization services (OR = 1.457) than those enrolled in other health insurances or uninsured. Need factors had impacts on unwell migrants’ unmet healthcare needs. Other factors, including age and health behavior, were also found to significantly affect unwell migrants’ unmet health service needs. Specific gaps continue to exist between unwell migrants and non-migrants regarding the accessibility of local health services. Flexible policies, such as enhancing the health awareness of migrants and eliminating obstacles for migrants to access medical services, should be implemented to provide convenient and affordable healthcare services to unwell migrants.

## 1. Introduction

In recent decades, continuous economic and social development created favorable conditions for global migration, rapidly expanding the scale of the global migrant population [1]. In China, internal migration is the predominant form of migration. The population of migrants rose from 221 million to 253 million during 2011–2014, representing more than one-sixth of the national population [2,3]. China’s interprovincial migration intensity has strong spatial variations, and the migration intensity has continued to increase in eastern China. Eight key migration destination provinces are founded and most of them are located in the eastern China area, such as the Pearl River Delta, Yangtze River Delta, and Beijing–Tianjin–Hebei Metropolitan Region. On the contrary, the central and southwestern parts of China have high out-migration probabilities and low in-migration probabilities, and the in-migration probabilities are quite stable while the out-migration probabilities are increasing in these regions. Moreover, internal migration is much more active in South China when compared to North China in the past several decades. The major migrant destinations and economic centers always overlap, showing that migration is mainly caused by economic agglomeration. Additionally, it could be concluded that the spatial-temporal pattern of China’s internal migration is consistent with Fujita’s “Core and Periphery Theory” [4]. Female and elderly migrants keep increasing because of the growth of the tertiary industry and financial expenditure on culture and recreation [5].

As China’s largest economic center and an international port city, Shanghai is a major attraction for the migrant population. Since the early 1990s, the migrant population has become the main source of permanent population growth in Shanghai. Shanghai had a migrant population of 3.057 million in 2000. According to the national 1% population sample survey and Shanghai’s sixth census data, this number had reached 4.384 million and 8.977 million in 2005 and 2010, respectively [6]. The migrant population of 2010 has increased by 193.6% compared to the data of 2000, which is almost tripled from 2000 to 2010. Shanghai’s seventh census, which is the latest census conducted in 2021, indicated a population of migrants in the city of 10.480 million. The migrant population has increased by 16.7% when compared to the data of 2010. Although the growth of the migrant population was slower during the last ten years compared with 2000 to 2010, the proportion of the migrant population increased from 25.4% in 2010 to 42.1% in 2021. The migrant population has already become an important component of Shanghai [7]. Moreover, the number of elderly migrants has begun to increase. In 2017, the number of the migrant elderly population older than 65 years old in Shanghai reached 0.307 million. This number had increased by 35.0% compared to 2016 [8].

Although the process of migration may potentially promote economic development and maintain social stability, migrants are marginalized due to lack of household registers in their host cities [9]. In China, the household register is regarded as a domestic passport, which plays a significant role in every aspect of an individual’s life. The household register is not only used to regulate the distribution of domestic residents but also dictates the availability of local social welfare, including social medical insurance, housing, and education [9,10,11]. In addition, migrants’ circumstances are unstable due to frequent career transitions and lifestyle changes [12,13]. Compared with non-migrants, migrants are more likely to have less social support, possess low levels of education, income, and health risk awareness, and experience worse living and work environments. As a result of these factors, migrants are more vulnerable when facing health problems [14,15,16,17].

The goal of universal health coverage is to ensure that all people have access to equal and effective health services without financial burdens [18]. Thus, the aim of universal health coverage is to narrow the gaps in the accessibility and decrease financial risk of health care among different countries, regions, and population groups [19]. Universal health coverage ensures that people have equitable access to health services according to their demands, regardless of their economic status or social background [18]. In China, migrants play a significant role in the modernization process, and their health status and access to health care are now among the most concerning public health problems [9].

Unmet healthcare needs, which are often used as indicators of health care availability [20,21,22], are subjective feelings experienced when people cannot obtain the proper medical services they need. It is preferable to use unmet healthcare needs rather than healthcare utilization as an accurate indicator of health availability because the former is able to reflect the overall healthcare needs rather than the actual utilization of health services [23,24]. Due to limited accessibility or inaccessibility of healthcare services, unmet needs consist of the non-utilization of outpatient or inpatient services when they are required [24]. Migrants may not utilize medical services due to weak awareness of health, poor economic circumstances, cumbersome medical procedures, and tedious work [14]. Unmet healthcare needs are an urgent issue that must be addressed via the accessibility of health services. If these needs are not met, the lack of healthcare can not only lead to poorer health status for migrants but also damage health equity, which hinders the realization of universal health coverage [25].

Past studies explored health service utilization behavior and influencing factors among migrants. For example, a study conducted in Beijing indicated ethnicity and severity of symptoms were significant determinants of healthcare utilization behavior among migrants [9]. National research found that internal migrants were more inclined to go to primary health facilities for treatment compared with urban residents [1]. Another nationwide study showed that the structural social capital, including the participation in social activities, integration of social organizations, and neighborhood characteristics, were significantly related to the establishment of health records and the reception of health education among migrants [26]. Another study in Beijing showed that age, education level, health status, work intensity, and health-related behavior affected the utilization of health education among migrants [27]. Studies carried out in Spain and Norway found that immigrants from low-income countries attended general practice more frequently [28,29,30]. In the Netherlands, immigrants made less use of family doctor services because of their lower health literacy [31]. A German study noted that migrants’ greater use of primary health care was due to their lower level of education, socio-economic status, and household income, which made it difficult to access higher levels of health care [32]. A study conducted in the Netherlands showed that the chance of unplanned re-admission was higher for Turkish in the “older than 45 years old” age group than for nationals [33]. A study undertaken in Italy showed that immigrants from places such as Turkey and Morocco had a longer average length of stay when hospitalized, which might be partly due to socio-economic indicators [34]. A study in the Czech Republic found that the type of visa held by migrants affected their use of inpatient services, and immigrants with long-term visas made less use of hospital services than those with permanent visas [35]. Many studies have shown that immigrants had a significantly lower utilization of health screening services than non-migrants [36,37,38]. Migrants face obstacles in the use of non-emergency and screening services due to language barriers, lack of health literacy, and ignorance of the benefits of using health services [34,39]. Reasons for the low update of screening among immigrants include lack of education about proper screening practices, lack of health insurance, and poor economic circumstances [37,38,40]. Studies from Canada have shown that migrants often did not effectively use mental health services and support due to language barriers, inadequate financial and social support, and distrust of mental health services [41,42]. However, a survey conducted in London showed that there were few differences in health service use between migrants and non-migrants, although migrants have disadvantages in socio-economic status [43]. A Spanish study of female and male visits to specialists found that male immigrants visited specialists less often than non-migrants [44]. A study from Italy found that migrants with poorer health, less healthy behavior, or lower socio-economic status used emergency services more often than non-migrants [45]. Some studies suggested limited time and lack of awareness about the potential risks to their health were reasons why migrants failed to recognize illness and utilize health services [31,46].

Previous research has mainly focused on the healthcare utilization behavior of migrants in specific areas [47] or on only one type of unmet healthcare need [17]. Many scholars and policymakers have paid attention to the healthcare utilization behavior of migrants, but research on the relationship between various unmet needs of unwell migrants and their determining factors in China is lacking. Migrants who are unwell have health service needs, and thus need to be paid additional attention. Therefore, in this research, we utilized data from the National Health Service Surveys of Shanghai to explore the potential influencing factors of various unmet healthcare needs of unwell migrants in Shanghai.

The contributions of this paper are as follows: (1) the potential differences in health status reported by unwell migrants and non-migrants are appraised; (2) we describe the unmet healthcare needs of unwell migrants and non-migrants and evaluate the gap between the unmet healthcare needs of the two groups; (3) we reveal the main underlying determinants of non-utilization of healthcare in different settings of demography, social structure, health-related behavior, and health conditions for unwell non-migrants and migrants of Shanghai.

### Theoretical Framework

As a well-tested theoretical analysis framework, the Anderson health service utilization model is a reliable tool to explore the influences on health service utilization because it considers both individual and social factors [48]. In this model, healthcare utilization is decided by three motivations: predisposing, enabling, and need factors. Predisposing factors refer to the individual characteristics of people who tend to utilize healthcare services before getting sick, and usually include demographic characteristics (race, gender, age, etc.) and social structure characteristics (education level, occupation type, etc.). Enabling factors are resources that can indirectly support the adoption of healthcare, including personal resources, family resources, and social resources (income, health insurance, etc.). Need factors can most directly affect the utilization of healthcare and include actual needs and self-perceived needs (self-evaluation of health status, number of chronic diseases, etc.).

In this study, health-related behavior variables were integrated into a classic model [27], which was further used to direct our empirical work (see Figure 1).

## 2. Materials and Methods

### 2.1. Data Sources and Study Population

The data of the study sample were taken from the Sixth Health Service Survey conducted in Shanghai in 2018. The survey has been conducted by the Center for Health Statistics and Information of the Health Commission of China every five years since 1993. As an important source of information for the government to comprehend the health status of residents and utilization of health services, this cross-sectional survey can be used to investigate the effects of previous health policies by providing insights into the characteristics of the medical industry, and the residents’ health service needs and demands. This survey can provide important information for the formulation of health strategies and promote the health of residents.

The survey adopted the principle of multi-stage stratified random sampling, and 16 districts of Shanghai were investigated. A total of 5 streets were selected from each district; 2 neighborhood committees were selected from each street; 60 households were selected from each neighborhood committee. Furthermore, 1200 households were selected from Pudong New District due to its high proportion of the population, and 600 households were selected from every other district. Respondents were residents with Shanghai household registration and internal migrants who had lived in Shanghai for at least six months, while visitors to Shanghai were excluded. In addition, non-migrants and internal migrants who participated in the survey perceived themselves to be unwell. As shown in Figure 2, five dimensions were used to determine whether a respondent was unwell. In the first dimension, the respondent was diagnosed by a doctor as suffering from hypertension, diabetes, or other chronic diseases. In the second dimension, the respondent consulted a doctor because of discomfort. In the third dimension, the respondent had an online medical consultation. In the fourth dimension, the respondent had self-medication and treatment behavior due to discomfort. In the fifth dimension, the respondent took leave or bed rest for one day or more from work or school due to discomfort. In any one of these five situations, the respondent would be considered to be unwell. The total number of unwell people was 10,938, including 934 internal migrants (people registered with permanent residence outside Shanghai) and 10,004 non-migrants (residents registered with permanent residence in Shanghai).

### 2.2. Variable Selection

#### 2.2.1. Dependent Variables

The dependent variables were three different types of unmet medical needs: (1) non-utilization of outpatient services (regardless of whether outpatient treatment was received in the past two weeks); (2) self-medication (regardless of whether the respondent self-medicated in the past two weeks, where self-medication mainly includes purchasing medicines (herbal decoction products) directly from pharmacies without a doctor’s prescription due to discomfort, adopting medications already in the home or not given by a doctor, and receiving services in non-medical health institutions); and (3) non-utilization of inpatient services (regardless of whether the respondent underwent hospitalization during the past year).

#### 2.2.2. Independent Variables

The factors that affect residents’ unmet healthcare needs are independent variables. According to the Anderson health service utilization model, these independent variables are divided into predisposing factors, enabling factors, need factors, and health behavior [49]. Predisposing factors contain characteristics of demography and social structure: (1) gender (male, female); (2) family size (2 people or less, 3 people or more); (3) age (0–30 years, 30–45 years, 45–60 years, and over 60 years); (4) marital status (married, not married); (5) education level (junior high school and below, high school and junior college, bachelor degree and above); (6) employment (employed, unemployed, retired, and other); and (7) occupation category. There are three types of careers: professional and semi-professional work, manual work, and others. Professional and semi-professional positions refer to heads of institutions, professional and technical staff, clerical staff, and service workers. People engaged in manual work include those working in agriculture, forestry, fishery, and water resources, production staff, transport equipment personnel, and those engaged in other work.

The enabling factors are related to financial and other resources that can promote the utilization of health services [19], and they are: (1) medical insurance, which consists of Urban Employee Basic Medical Insurance, other, and uninsured (other insurances are Urban Resident Basic Medical Insurance, the New Rural Cooperative Medical Scheme, etc.); (2) average annual household income (less than CNY 100,000, CNY 100,000–300,000, CNY 300,000–500,000, more than CNY 500,000); (3) house ownership (own house, rent); (4) residence (rural, non-rural). The former refers to households of an agricultural nature, whereas the latter refers to households of a non-agricultural nature.

Need factors include subjectively perceived needs and objectively existing needs as judged by doctors [50], as follows: (1) anxiety, with two responses—yes (refers to a feeling of moderate or extreme anxiety today in terms of physical functioning), no (refers to unperceived anxiety today in terms of bodily functions); (2) self-reported health status (excellent, good, fair, poor); (3) the number of chronic diseases (no chronic illnesses, a chronic illness, multiple chronic illnesses).

Finally, health behaviors include smoking, alcohol consumption, and physical activity: (1) smoking (yes, no) is defined as having smoked a cumulative total of 100 cigarettes since the first cigarette and continuing to do at present, and non-smokers are those who have never smoked or have not smoked a cumulative total of 100 cigarettes; (2) drinking (yes, no) refers to having consumed alcohol in the last 12 months; (3) conducting physical exercise is measured as never, 1–2 times a week, or more than 3 times a week.

### 2.3. Statistical Analysis

#### 2.3.1. Descriptive Analysis

Descriptive analysis methods were applied to describe the characteristics of the sample population, including general characteristics, trends in health status, and conditions of unmet health needs among unwell migrants and non-migrants.

#### 2.3.2. Binary Logistic Regression Analysis

The study estimated the relationship between unmet health service needs and various influencing factors in the Anderson model using binary logistic regression. The logistic regression contained two models, namely, model I for migrants and model II for non-migrants. The dependent variables in the model were categorized into two classes, namely, the utilization of health services and the non-utilization of health services (self-treatment, non-utilization of physical examinations, non-utilization of inpatient services, and non-utilization of outpatient services). We used SPSS 24.0 (SPSS Inc., Chicago, IL, USA) for Windows to analyze the data. The chi-square test and logistic regression were used to examine or analyze differences and influencing factors, respectively. If *p* < 0.05, the result was considered to be statistically significant.

### 2.4. Ethical Considerations

The Health Service Survey was organized and led by China’s National Health Commission. The Planning Department and the Statistical Information Center of the Commission were responsible for the coordination and implementation, respectively. Prior to the survey, the investigator notified respondents of a statement involving the purpose of the survey and the confidentiality of respondents’ responses. The survey emphasized the principle of voluntary participation, i.e., the respondent had the right to choose not to take the survey. The study was approved by the Ethics Committee of the School of Public Health of Fudan University.

## 3. Results

### 3.1. Sample Characteristics

The study surveyed 109,838 respondents with health service needs. Elderly respondents (age > 60) comprised 67.99% of the total; other demographic and socio-economic characteristics of the interviewees are presented in Table 1. The enabling factors for unwell migrants and non-migrants were different. Migrants are disadvantaged in terms of certain social resources. The participation rate in Urban Employee Basic Medical Insurance for unwell migrants was low (51.96%) compared with unwell non-migrants (59.10%). In addition, the percentage of non-migrants (94.13%) with housing ownership was more than that of migrants (60.60%) with housing ownership. Regarding need factors and health behavior, migrants had better health status but showed poorer health behavior. Non-migrants (48.66%) were more likely to undertake regular physical activity (more than three times per week) than migrants (46.24%), suggesting non-migrants paid more attention to exercise.

### 3.2. Health Status and Unmet Healthcare Needs

A total of 197 unwell migrants (21.09%) and 2835 unwell non-migrants (28.34%) reported that their health status was fair or poor (see Table 2). The results showed that 681 unwell migrants (72.91%) and 8868 unwell non-migrants (88.64%) suffered from at least one chronic disease. In different age groups, the trends of people suffering from chronic disease among unwell migrants and non-migrants were different. In the 30–45 and 45–60 age groups, the percentage of unwell non-migrants suffering from chronic disease was lower than that of unwell migrants, but the difference in other age groups was not statistically significant (see Table 2).

As shown in Table 3, the total percentages of non-utilization of inpatient service and outpatient service were 86.46% and 38.51%, respectively. The rates of non-utilization of these two medical and health services of migrants were higher than those of non-migrants. In addition, the percentage of unwell migrants taking self-treatment (23.98%) was higher than that of unwell non-migrants (17.97%).

### 3.3. Logistic Regression for Main Reasons for Unmet Healthcare Needs

The data presented in Table 4 and Table 5 show the results of the binary logistic regression for migrants and non-migrants not utilizing health services, respectively.

Regarding predisposing factors, younger migrants were significantly more likely to self-medicate (OR = 2.687) and not utilize inpatient services (OR = 3.209) than older people aged over 60. However, persons aged under 60 indicated a significantly higher probability of self-medicating (≤30: OR = 1.669, 45–60: OR = 2.054) and use of outpatient services (30–45: OR = 1.435, 45–60: OR = 1.153) among non-migrants. Non-migrants living in smaller families (households ≤2) indicated a significantly lower probability of not utilizing outpatient services (OR = 0.818), but no significant effect of family size was found in migrants. Similarly, the unmarried demonstrated a significantly higher probability of self-treatment (OR = 1.205) among non-migrants, but marriage status had no significant effect on the unmet health service needs of migrants.

Regarding enabling factors, unwell non-migrants from rural areas were significantly less likely not to utilize inpatient services (OR = 0.845), and more likely to utilize self-treatment (OR = 1.393). In addition, it should be noted that, compared with those enrolled in Urban Employee Basic Medical Insurance, unwell migrants enrolled in other health insurance systems and those who were uninsured indicated a significantly higher probability of not utilizing outpatient services (OR = 1.457), but there was no significant effect in terms of inpatient service utilization (*p* > 0.05). Non-migrants with other health insurance and the uninsured showed a significantly higher probability of self-medicating (OR = 1.214), compared with non-migrants enrolled in Urban Employee Basic Medical Insurance.

Regarding need factors, subjective health status (self-evaluated health status) reported by respondents and objective health status (suffering from chronic disease) diagnosed by doctors had significant effects on the non-utilization of medical and health services among migrants and non-migrants. First, migrants with better self-reported health status (excellent: OR = 5.070, good: OR = 4.359, fair: OR = 3.139) showed a significantly higher probability of not utilizing inpatient services than those with worse self-reported health status. The same effect was seen in non-migrants (excellent: OR = 3.095, good: OR = 2.130, fair: OR = 1.301). Second, migrants and non-migrants with one chronic disease indicated a significantly lower probability of not utilizing inpatient services than those without the chronic disease (OR = 0.450, 0.542). Migrants and non-migrants with one chronic disease showed a lower probability of self-treatment than those without the chronic disease (OR = 0.321, 0.493), and the same phenomenon was found in patients with multiple chronic diseases (OR = 0.365, 0.468). On the contrary, patients with multiple chronic diseases were more likely to utilize outpatient services among migrants and non-migrants (OR = 2.212, 2.373).

In terms of health behavior, migrants with smoking habits were significantly more likely not to utilize outpatient services (OR = 1.703), and non-migrants with drinking or smoking habits were significantly more likely not to utilize healthcare services (inpatient services: OR = 1.455, outpatient services: OR = 1.145).

## 4. Discussion

Mitigating unmet healthcare needs plays a significant role in reducing health inequality and fulfilling the goal of universal health coverage [51].

Although many studies have focused on the healthcare utilization of migrants, research on the various unmet healthcare needs of unwell migrants and their determinants is scarce. In this paper, we presented the social and economic status, health status and behavior, and unmet healthcare needs of unwell migrants and non-migrants and compared the situation of the two groups. In addition, we explored the potential effects of certain influencing factors on unmet healthcare needs among unwell migrants and non-migrants. The main purpose of this paper was to explore those factors’ influences on the unmet health service needs of unwell migrants based on an understanding of their status and to provide relevant recommendations for reducing these unmet health needs.

### 4.1. Migration Status and Unmet Health Needs

#### 4.1.1. Social and Economic Resources

Migrants are somewhat disadvantaged in terms of their social and economic resources. We found that unwell migrants who were enrolled in other health insurance or were uninsured were more likely to not utilize medical services than those enrolled in the Urban Employee Basic Medical Insurance. This result is consistent with previous research findings in China [1]. Only migrants with formal jobs can join the Urban Employee Basic Medical Insurance in their host cities. Although informally employed migrant workers can chose to join Urban Resident Basic Medical Insurance and the New Rural Cooperative Medical Scheme in their registered place of residence or not to participate in any health insurance [52], other forms of basic medical insurance have poorer health care coverage and lower reimbursement rates than Urban Employee Basic Medical Insurance in Shanghai, e.g., the reimbursement rate of the New Rural Cooperative Medical Scheme was less than that of Urban Employee Basic Medical Insurance [53]. Furthermore, other forms of health insurance for migrants are based on their domicile registration locations; thus, in addition to rarely covering medical expenses outside of the place of origin, these schemes also have complex reimbursement processes [54]. Migrants also have a lower rate of homeownership than non-migrants.

#### 4.1.2. Health Status and Behavior

Our research found that the proportion of unwell non-migrants with a fair or poor self-evaluated health status and suffering from chronic diseases was significantly higher than that of unwell migrants, regardless of age. This discovery confirmed the theory that migrants have health advantages compared to non-migrants [55]. Another study also indicated that migrants were healthier than non-migrants, which is known as the healthy migrant effect [56]. Health advantages were especially reflected in mortality, and migrants who entered the host city for a short time were generally healthier [57]. Similar to previous research conclusions [58,59], our research also found that the health advantages of unwell migrants diminished particularly sharply in middle age because they usually faced many pressures during this period [59].

In addition, we also found that unwell migrants had poorer health-related behavior than unwell non-migrants. Research has shown that health behavior and health awareness are closely related [60]. An effective measure to change health behavior is to improve the health literacy of the population in community settings [61]. To mitigate the risky habits of migrants, health promotion efforts should be accompanied by educational interventions to strengthen health literacy.

#### 4.1.3. Unmet Health Needs

In general, the degrees of self-treatment, non-utilization of outpatient service, and non-utilization of inpatient services among unwell migrants and non-migrants were significantly different. Unwell migrants had more unmet healthcare needs than unwell non-migrants, and this result was in accordance with previous surveys in China [1,62,63,64]. This may be due to migrants’ inadequate comprehension of health caused by their poor working and living environments [65]. We have found similar phenomena in Italy, where migrants had challenges in accessing healthcare services after migrating to the host country [45]. Additionally, migrants were inclined to ignore health issues, which results in more unmet healthcare needs [9,31]. Therefore, policy efforts should be concentrated on improving the socio-economic environment and health awareness of migrants, and those migrants who are unwell must be given priority attention.

### 4.2. Influencing Factors of Unmet Healthcare Needs

#### 4.2.1. Enabling Factors

Previous research indicated that participating in medical insurance could significantly promote the utilization of healthcare services [66]. We found that participation in Urban Employee Basic Medical Insurance reduced unwell migrants’ unmet needs for outpatient services but unmet needs for inpatient services were not addressed, which was consistent with other findings in Shaanxi [67]. Studies in Spain have also shown that lack of health insurance discouraged migrants from using health screening services [37].

Residents enrolled in Urban Employee Basic Medical Insurance had a significant advantage in utilizing medical resources compared with those covered by other health insurance [68]. The facilitating role in health service utilization showed differences between health insurance systems, and Urban Employee Basic Medical Insurance indicated a relatively stronger impact than other medical insurance. This may be due to the fact that the Urban Employee Basic Medical Insurance has an advantage compared to Urban Resident Basic Medical Insurance and the New Rural Cooperative Medical Scheme in terms of their reimbursement policy, which have the lowest out-of-pocket proportions [69]. In addition, migrants enrolled in the resident health insurance are required to return to their place of residence to finish the reimbursement procedure, which may limit migrants’ motivation to care for their health [13].

The unmet needs of unwell migrants who are enrolled in other medical insurance schemes, or are uninsured, are a critical issue that must be addressed [1]. Due to the effect of household registration, it is a challenge for migrants to obtain access to local social welfare. European migrants face similar barriers in accessing healthcare services. Although the human right to health was enshrined in the Constitution of the World Health Organization in 1948, there has been a failure to accommodate diversity in the health service systems of host countries [70]. The link between household registration and welfare entitlements still exists, and health insurance is based on household registration [71]. The uneven economic development in various regions of China is accompanied by an unequal allocation of health and welfare benefits [72]. In most developed areas, the social welfare system established for migrants is usually different from non-migrants [73]. In 2009, a new round of health care system reform was initiated by the Chinese government to achieve the goal of universal health coverage [74]. Since then, a comprehensive range of health reforms has been gradually adopted, including the expansion of health insurance and the reform of public hospitals [75]. As of 2018, the number of participants in China’s basic medical insurance reached 1.34 billion, with a participation rate of over 95% [76]. However, a unified pool of health insurance funds at the national level has not yet been established in China. In most areas, migrants enrolled in the New Rural Cooperative Medical Scheme are unable to directly access local medical insurance, potentially hindering their utilization of medical services [77]. In addition, as non-permanent residents of host cities, the health insurance of migrants is typically based on the individual’s home city, and migrants usually must return to their home city for reimbursement [13]. An intricate reimbursement process and a lower proportion of financial compensation for migrants, compared to locals, may reduce the probability of migrants utilizing medical services [13]. China launched insurance for migrant workers in 2006, but its coverage was limited to migrants with stable employment, and self-employed migrants or those without work contracts were excluded [9]. To address the above issues, we believe that the policies should be more flexible for migrants. Regarding the lack of coordinated policies for health insurance funds in China [78], it is preferable to increase the number of hospitals with the ability to settle medical bills on site. In addition, to promote equitable access to medical services for non-migrants and migrants, the Yangtze River Delta region is exploring the integration of medical insurance to reduce the differences between medical insurance schemes in different regions [79].

#### 4.2.2. Need Factors

In this study, we found need factors had significant impacts on unwell migrants’ non-utilization of hospitalization, non-utilization of outpatient care, and self-medication. Unwell migrants with poor self-evaluated health status had a smaller probability of non-utilization of medical services, which is consistent with previous surveys in Beijing [9].

We also found that unwell migrants tended to underutilize healthcare services prior to being diagnosed with chronic diseases [80]. Several reasons may explain the above differences. Firstly, due to the pressures of working and maintaining their livelihood, migrants usually do not want to take work leave to utilize healthcare services because they are concerned with the reduction in their wages [81]. Secondly, migrants may be more likely to ignore health risks and take self-medication, rather than utilizing healthcare services in medical institutions, than non-migrants [16]. Finally, before becoming permanent residents of the host city, social welfare is limited for migrants without formal employment [10]; therefore, migrants will return to their home city for treatment [82]. Regarding the issue of migrants choosing to utilize healthcare services only when they are seriously ill, we offer two suggestions. First, narrowing the gaps between non-migrants and migrants in terms of access to housing, education, and public health services, and improving the accessibility of medical services for migrants, would be effective means to reduce their unmet healthcare needs. Second, it would also be effective for companies to provide employees with health promotion materials, health consultations, and physical examinations, which lead employees to pay more attention to their health.

#### 4.2.3. Predisposing Factors and Health Behavior

Other factors that affect the unmet healthcare needs of migrants are age and other health-related behavior. Specifically, the elderly showed a lower probability of abandoning inpatient services and performing self-medicine [80]. The high risk of disease for older people, including chronic diseases and disabilities, results in a greater need for healthcare services among elderly migrants [83]. We also found that migrants with a smoking habit were more likely to abandon healthcare services. Respondents’ health literacy is connected to the population’s health knowledge, health motivations, health-related behavior, and health outcomes [84]. Thus, the government can raise the health awareness of residents by enhancing health literacy, which may alleviate unmet healthcare needs [19].

We acknowledge that this study has the following limitations. First, although our survey was conducted among residents who had been living in Shanghai for more than six months, the important potential variable of the duration of the migration was not considered. Second, our study did not analyze the changing trends of migrants’ unmet healthcare needs over time. Due to the difficulty in obtaining the original data of the National Health Service Survey in Shanghai, our study only discussed the unmet health service needs of migrants in 2018. In the future, we will use data and variables from other surveys to reveal the changing trends of unmet healthcare needs of migrants on migrant healthcare utilization, as well as the impact of the duration of migration, which may lead to other meaningful conclusions.

## 5. Conclusions

The findings of this study will contribute to the reform of the healthcare system. Health reform policies that emphasize equity in health outcomes will focus more on internal migration. The formulation of policy should focus on improving the health literacy of migrants and eliminating obstacles for migrants to access medical and healthcare resources. This means the focus in the future should be on improving health education for migrants in the workplace, promoting the homogenization of health insurance between different regions, and narrowing the gap between non-migrants and migrants in terms of access to housing, education, and public health services. The implementation of the above measures will help to improve the social integration of migrants in the host city and lower the barriers to access to health care services for vulnerable groups such as unwell migrants.

Although the data of this survey were mainly collected in Shanghai, the results of this research may also be meaningful for other cities in China or other countries, particularly developing countries. In most developing countries, economic development is uneven, and migrants tend to migrate to areas with higher economic levels. A concomitant problem is that migrants have barriers to accessing health services in the host city or country. Despite the differences in the regions studied, our study reached similar conclusions to other studies. Studies from China, Italy, and the US have shown that unwell migrants have better health status and more unmet healthcare needs than non-migrants. Studies on China as a whole, Beijing, and the US have found that the need factors, including self-assessed health status and the impact of chronic diseases, are the main determinants of unmet healthcare needs among migrants. Studies on China as a whole, Guangzhou, Shaanxi, Beijing, Spain, and the US have indicated that medical insurance and health behavior may also have a significant impact on the unmet healthcare needs of migrants. Additionally, other studies in China have also found that age is significantly related to migrants’ unmet healthcare needs. Therefore, we believe that our findings can, to some extent, provide partial lessons for other regions, and that the results of this survey may be of significant importance as a reference for other cities or countries to satisfy unmet healthcare needs for migrants and improve social fairness.

In the future, more comprehensive and in-depth research is needed to examine how the unmet health service needs of migrants change with the migration duration in the host city, to thus improve migrants’ access to health services. In addition, survey data from different years can be used to explore changing trends in the unmet health needs of migrants and the factors that influence them. It is proposed to use a time-varying difference-in-difference approach and event study method to assess the impact of health policies targeting migrants during the study period.

## Figures and Tables

**Figure 1 ijerph-19-05499-f001:**
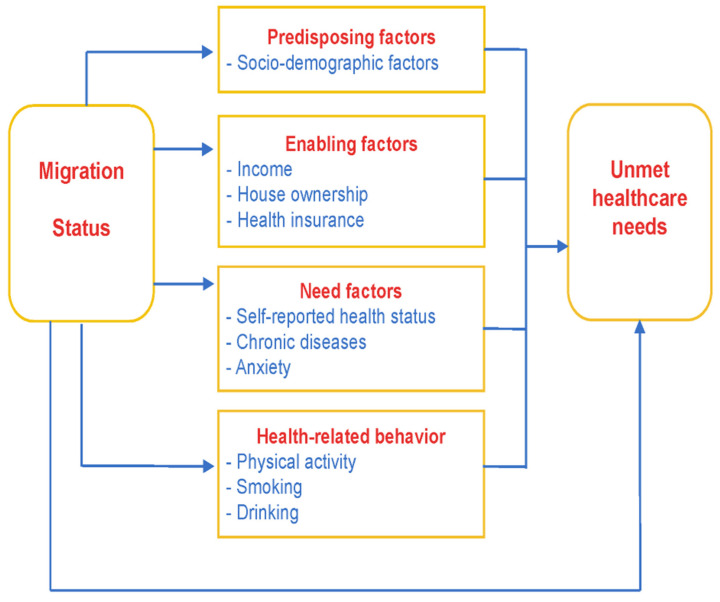
The simplified Anderson health behavioral model.

**Figure 2 ijerph-19-05499-f002:**
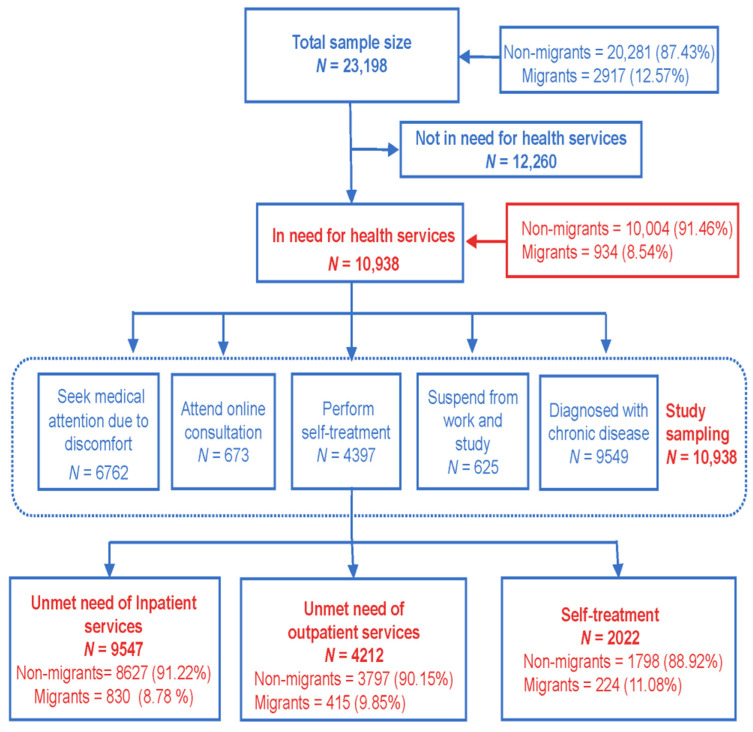
The sample size of the study and their unmet health service needs.

**Table 1 ijerph-19-05499-t001:** Characteristics of unwell migrants and non-migrant participants in Shanghai.

	Variable		Total *n* = 10,938		*p*-Value
			Migrants *n* = 934	Non-Migrants *n* = 10,004	
Predisposing factors	Gender	Male	45.72%	47.08%	0.424
		Female	54.28%	52.92%	
	Age	≤30	11.67%	2.71%	0.000
		31–45	15.85%	5.12%	
		46–60	25.27%	22.24%	
		>60	47.22%	69.93%	
	Family size	≤2 people	74.73%	45.67%	0.000
		≥3 people	25.27%	54.33%	
	Marital status	Unmarried	16.70%	18.88%	0.102
		Married	83.30%	81.12%	
	Education level	Junior high school and below	44.22%	63.50%	0.000
		High school and junior college	41.22%	29.86%	
		Bachelor’s degree and above	14.56%	6.64%	
	Employment	Employed	33.83%	17.10%	0.000
		Unemployed	12.10%	13.62%	
		Retirement and other	54.07%	69.27%	
	Occupation category	Professional and semi-professional work	52.57%	3.94%	0.000
		Manual work	25.59%	81.12%	
		Other	21.84%	14.94%	
Enabling factors	Income (CNY)	<100,000	44.22%	60.82%	0.000
		100,000–300,000	48.50%	36.43%	
		300,000–500,000	5.25%	2.27%	
		>500,000	2.03%	0.49%	
	House ownership	Own house	60.60%	94.13%	0.000
		Rent	39.40%	5.87%	
	Residence	Non-rural	24.63%	21.06%	0.011
		Rural	75.37%	78.94%	
	Medical insurance	Urban Employee Basic Medical Insurance	51.96%	59.10%	0.000
		Other and uninsured	48.04%	40.90%	
Need factors	Self-reported health status	Excellent	55.46%	47.98%	0.000
		Good	23.45%	23.68%	
		Fair	18.84%	25.10%	
		Poor	2.25%	3.24%	
	Chronic diseases	No chronic	27.09%	11.36%	0.000
		One chronic	48.72%	55.80%	
		Multiple chronic	24.20%	32.85%	
	Anxiety	Yes	6.64%	8.37%	0.066
		No	93.36%	91.43%	
Health behavior	Smoking	Yes	18.20%	19.44%	0.358
		No	81.80%	80.56%	
	Drinking	Yes	29.44%	22.43%	0.000
		No	70.56%	77.57%	
	Physical activity	Never	4.50%	1.15%	0.000
		1–2 times a week	49.25%	50.19%	
		More than 3 times a week	46.25%	48.66%	

**Table 2 ijerph-19-05499-t002:** The health status of unwell migrants and non-migrants in different age groups.

	Total		Self-Evaluation of Health Status as Fair and Poor			Suffering from Chronic Diseases		
	Migrants	Non-Migrants	Migrants	Non-Migrants	*p*-Value	Migrants	Non-Migrants	*p*-Value
≤30	109 (11.67%)	271 (2.71%)	3 (1.52%)	13 (0.46%)	0.369	20 (2.94%)	50 (0.56%)	0.982
30–45	148 (15.85%)	512 (5.12%)	13 (6.60%)	63 (2.22%)	0.237	64 (9.40%)	341 (3.85%)	0
45–60	236 (25.27%)	2225 (22.24%)	52 (26.40%)	475 (16.75%)	0.807	190 (27.90%)	191 (21.54%)	0.028
60	441 (47.22%)	6996 (69.93%)	129 (65.48%)	2284 (80.56%)	0.140	407 (59.77%)	6567 (81.52%)	0.184
Total	934	10004	197 (21.09%)	2835 (28.34%)	0	681 (72.91%)	8868 (88.64%)	0

**Table 3 ijerph-19-05499-t003:** The prevalence of unmet healthcare needs of unwell migrants and non-migrants in Shanghai.

	Total (*n* = 10,938)	Migrants (*n* = 934)	Non-Migrants (*n* = 10,004)	*p*-Value
Non-utilization of inpatient services	9457 (86.46%)	830 (88.87%)	8627 (86.24%)	0.025
Non-utilization of outpatient service	4212 (38.51%)	415 (44.43%)	3797 (37.95%)	0.000
Self-treatment	2022 (18.49%)	224 (23.98%)	1798 (17.97%)	0.000

**Table 4 ijerph-19-05499-t004:** Logistic regression of unmet healthcare needs among unwell migrants in Shanghai.

Variable	Non-Utilization of Inpatient Services		Self-Treatment		Non-Utilization of Outpatient Services	
	OR	SD	OR	SD	OR	SD
Predisposing factors						
Age (Ref: >60)						
≤30	1.213	0.675	2.687 *	0.455	1.817	0.399
30–45	1.097	0.577	1.962	0.401	1.527	0.337
45–60	3.209 *	0.457	1.208	0.288	1.430	0.234
Medical insurance (Ref: Urban Employee Basic Medical Insurance)	1.497	0.275	1.292	0.204	1.457*	0.173
Need factors						
Self-reported health status (Ref: Poor)						
Excellent	5.070 **	0.541	0.715	0.602	1.301	0.528
Good	4.359 **	0.555	0.697	0.617	1.224	0.539
Fair	3.139 *	0.535	0.824	0.614	0.918	0.538
Chronic diseases (Ref: No chronic)						
One chronic	0.450 *	0.400	0.321 ***	0.292	0.926	0.256
Multiple chronic	1.084	0.349	0.365 ***	0.217	2.212 ***	0.203
Health behavior						
Smoking (Ref: No)	1.007	0.350	1.119	0.250	1.703 *	0.217

*** Represents *p* < 0.001, ** represents *p* < 0.01, * represents *p* < 0.05; OR represents odd ratio; SD represents standard deviation.

**Table 5 ijerph-19-05499-t005:** Logistic regression of unmet healthcare needs among unwell non-migrants in Shanghai.

Variable	Non-Utilization of Inpatient Services		Self-Treatment		Non-Utilization of Outpatient Services	
	OR	SD	OR	SD	OR	SD
Predisposing factors						
Age (Ref: >60)						
≤30	2.360	0.440	1.786 **	0.217	1.427	0.191
30–45	0.958	0.197	1.889 ***	0.150	1.435 **	0.121
45–60	1.206	0.096	1.205 *	0.079	1.153 *	0.063
Family size (Ref: ≥3)	0.899	0.064	0.903	0.057	0.818 *	0.046
Marital status (Ref: Married)	1.091	0.080	1.205 **	0.071	1.038	0.058
Employment (Ref: Employed)						
Unemployed	0.813	0.132	1.359 **	0.104	0.833 *	0.082
Retirement and other	0.656 *	0.175	0.562 ***	0.149	0.889	0.116
Occupation category (Ref: Professional and semi-professional work)						
Manual work	1.044	0.137	1.121	0.112	1.043	0.091
Other	0.999	0.078	1.211 **	0.068	1.051	0.054
House ownership (Ref: Rent)	0.948	0.137	1.341 *	0.129	1.01	0.095
Residence (Ref: Non-rural)	0.845 *	0.084	1.393 ***	0.074	0.891	0.062
Medical insurance						
(Ref: Urban Employee Basic Medical Insurance)	0.891	0.070	1.214 **	0.061	0.987	0.049
Need factors						
Self-reported health status (Ref: Poor)						
Excellent	3.095 ***	0.141	0.894	0.153	1.194	0.131
Good	2.130 ***	0.143	0.828	0.156	1.016	0.133
Fair	1.301 *	0.133	0.891	0.152	0.957	0.130
Chronic disease (Ref: No chronic)						
One chronic	0.542 ***	0.128	0.493 ***	0.091	1.256 **	0.086
Multiple chronic	0.898	0.126	0.468 ***	0.083	2.373 ***	0.079
Anxiety (Ref: Yes)	1.245 *	0.095	0.892	0.097	1.123	0.082
Health behavior						
Smoking (Ref: No)	1.455 ***	0.097	1.023	0.083	1.088	0.065
Drinking (Ref: No)	1.112	0.088	1.081	0.076	1.145 *	0.059
Physical activity (Ref: More than 3 times a week)						
Never	0.776	0.562	1.324	0.298	0.904	0.278
1–2 times a week	1.156 *	0.063	0.979	0.056	0.835 ***	0.044

*** Represents *p* < 0.001, ** represents *p* < 0.01, * represents *p* < 0.05; OR represents odd ratio; SD represents standard deviation.

## Data Availability

The data will not be shared. All data used in this paper were supported by the municipal governments of Shanghai, China. In order to protect residents’ privacy, the municipal governments did not give us the authority to make the data public.

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
