# Peer review of "Unmet Healthcare Needs and Their Determining Factors among Unwell Migrants: A Comparative Study in Shanghai"

_ijerph, 2022, doi:10.3390/ijerph19095499_

Round 1

Reviewer 1 Report

The paper has been revised and resubmitted, as far as I can see. Even though I quite liked the first iteration of the paper, I can see some further improvements in this revised version. Nevertheless, I think that the paper can be further polished and improved. Please see my comments below:

  1. There was not much added about immigrants in China and in Shanghai specifically. It should be explained whether they domestic (rural-urban) migrants or are they international migrants? What are the basic statistics of these people? How does migration evolve in China? An additional 1-2 pages can be spent describing that background.
  2. The Conclusions still need to be further extended to include the policy implications, a longer elaboration on the pathways for further research, but most important of all, the applicability of results and their universal importance for other countries that might experience similar issues.
  3. The paper would still benefit from another round of a thorough English proofreading.

Reviewer 2 Report

I have nothing further to add to the revisions - I look forward to seeing this article in print and citing it in my own work. 

Reviewer 3 Report

I appreciate the efforts of the authors to further develop this paper.  However, the paper remains fatally flawed.  The authors note in their statement of limitations, “First, although our survey was conducted among residents who had been living in Shanghai for more than six months, the important potential variable of the duration of the migration was not considered. Second, the unmet healthcare needs were limited to residents who accessed healthcare services, but those who did not access healthcare services were excluded.” 

Not measuring duration of residence is a flaw, not a limitation.

Not including those who did not access healthcare services in a study of unmet healthcare needs is a flaw, not a limitation.  Those who had the least utilization of healthcare services (i.e., unmet healthcare needs) are not included in the study.

Round 2

Reviewer 3 Report

The analysis remains flawed -- no measure of length of residence -- but this cannot be addressed.

This manuscript is a resubmission of an earlier submission. The following is a list of the peer review reports and author responses from that submission.

Round 1

Reviewer 1 Report

The paper entitled "Unmet Healthcare Needs and Their Determining Factors among Migrants: A Cross Sectional Study in Shanghai" focuses on the analysis of the health status and unmet healthcare needs of migrants and explores the impact of related factors on migrants’ unmet healthcare needs in Shanghai, China. The paper has a clear structure and design. It employs robust tools and yields non-trivial results. I enjoyed reading the paper and I have several suggestions and recommendation for the authors:

  1. It should be clearly stated how the results from Shanghai regarding the unmet healthcare of migrants can be used in other countries. Are the results from this study universal or not?
  2. It would be interesting to learn more about immigrants in China and in Shanghai specifically. Are they domestic (rural-urban) migrants or are they international migrants? What are the basic statistics of these people? How does migration evolve in China?
  3. The paper would benefit from a longer literature review (some 10-15 additional sources can be included, or even more). Comparison with other similar cases around the world might be needed.
  4. The Conclusions need to be extended to include the policy implications, pathways for further research, but most important of all, the applicability of results and their universal importance. 
  5. The paper need a thorough English proofreading.

Reviewer 2 Report

Thank you for the opportunity to review this paper. It is an important contribution to the literature of migrants and their health needs. I have a few comments on the paper: 

Title: I suggest that the title of the paper reflect that this was a comparison between migrants and native population: A comparative study in Shanghai, so instead of a cross sectional study. The fact that it is cross-sectional can be made clear in the methods section of the paper.

Line 21 – 22: The authors make reference to the efficacy of health insurance in reducing migrants’ unmet health care needs has not been discovered. Could you please explain what is meant by “has not been discovered”?

Line 33 & 37: The authors make reference to “floating population”, and in terms of the migrants and migration literature this is not a term I am familiar with or have come across before. What I do know is that it suggests a cavalier approach to mobile populations, so I would suggest using nomenclature that better describes mobile populations.

Line 52 – 53: The sentence, “The goal of universal health coverage…” must be appropriately referenced/cited.

Line 64 – 67: The authors introduce the concept unmet needs as indicated by 3 factors of “non seek”. I wanted to find out whether this was how it appeared in the survey. If not, I would suggest using “did not seek”, i.e. did not seek outpatient services, did not seek inpatient services, and did not seek physical examination.

Line 72: UCH = UHC

Line 82: researches = research

Study population: Were these international or internal migrants? When the authors set the scene in the introduction, there was a summary of the global picture on migration, and then localized to Shanghai China. However, much later in the discussion, the authors refer to rural migrants (line 376). How was migrant defined in the SHS survey?

Figure 2: Perhaps it may be useful to add the proportion of participants, i.e. migrants and natives as part of Figure 2?

Table 1: Insurance = medical insurance

Line 190: How are mental and semi-mental workers defined? What work is involved with those participants who work in mental and semi-mental sectors? Similarly, are physical workers those participants who undertake manual labour?

Line 288: This is known as the healthy migrant effect (see literature on this).

Line 299: “…migrants are so optimistic…” what is meant by optimistic in relation to more unmet health care needs?

Line 344-347: Does the separate welfare system have limitations or restrictions for migrants? Additionally, national health insurance systems are the vehicle for achieving the goal of UHC. The authors state that this has not yet been achieved in China, so what systems have China begun to put into place to progress towards UHC?

I do think that the paper will benefit from editing – I picked up a few errors throughout the paper.

Reviewer 3 Report

This analysis addresses important issues of migrant health and health care utilization.  However, it should be substantially revised so that it can be given a fair review.

The entire paper must be edited for language, grammar, sentence structure, and word usage. For example, the authors use “mental” when they mean “menial;” they use “possibilities” when they mean probabilities.  They also inconsistently use terms; for example, they use the term “non-seek” (which make little sense in English) in the text, and the use the term “non-use” in the tables.  Not are non-seek and non-use different terms, but they convey very different meaning.

In addition to basic grammar and word usage, the authors should be attentive to structure.  For example, the paragraph from lines 52-89 addresses three ideas and should be separated into three paragraphs.

The authors need to provide a statement noting the ethical review of the research, including some statement about how consent was obtained.

The authors need to clarify the size of the sample used in the analysis.  In line 127 they refer to 23,198 respondents, on line 186 they refer to 109,838 respondents, and in table 1 they report results for 10,938 participants.

The authors need to provide better definitions for the variables used in the analysis.  Among the dependent variables, glosses such as “non-seek of physical examination” with a definition of whether the participant received a physical examination in the past 12 months is not clear.  Self-treated defined as whether someone self-medicated in the past two weeks leaves the reader to question whether self-medication includes vitamins, antiseptic cream, or aspirin.  Among other variables, anxiety as a simple dichotomous question must be clarified.  Similarly, what constitutes rural versus urban location?  The authors need to clarify if smoking refers to current behavior or behavior at any point in the participants life, as well as the amount; the same is true for drinking (assumed to mean alcohol consumption).

I question the value of the measure “insurance” in differentiating health care utilization.  98.56% have insurance of some kind.  It is little wonder that it is not associated with the outcomes.

I found the presentation of the tables and of the narrative results to be overwhelming.  Part of this can be addressed by editing for language.  However, the authors should consider providing greater structure to the presentation of results. 

The discussion appears to focus on the fact that migrants have better health and less utilization than do non-migrants.  This should not be surprising – the healthy migrate, the unhealthy remain at home. 

In reading the Discussion I also came to realize that important measures for understanding the results were not included in the analysis.  These include age at migration versus current age, and length of residence in Shanghai.  Without this information, it is impossible to interpret the results.

Round 2

Reviewer 3 Report

The authors have addressed many of my original concerns.  Unfortunately, this analysis remains fatally flawed.

As identified in my original review, the major fatal flaw is that the authors cannot address the length of time migrants have lived in Shanghai.  An immigrant who has resided in the city for 30 years will have better integrated than an immigrant who has resided in the city for 7 months.  The authors do not consider this in their analysis.  They do not even acknowledge this in their limitations.

A second fatal flaw is that the data used in the analysis does not address the purpose of the paper or its title.  The authors state on line 355 that, “The main purpose of this paper was mainly to further facilitate the healthcare utilization behavior of migrants in their host cities.” The title of the paper is “Unmet Healthcare Needs and Their Determining Factors 2 among Migrants: A comparative study in Shanghai.”  The authors include among their outcomes preventive health behaviors (I agree that preventive health behaviors are “healthcare needs”).  However, the data they use in their analysis is limited to persons who meet the inclusion criteria of natives and migrants who “perceived themselves to be unwell” (line 164).  This also results in a sample overwhelmingly lop-sided by age; it largely includes older adults, with few younger adults.

Finally, the entire paper requires further editing.  The use of contractions in a professional paper is not appropriate.  Stating that participants were “brain and semi-brain workers” (line 199) is inappropriate.
